



# Abiotic CO₂ sources confound interpretation of temperature responses of in situ respiration in geothermally warmed forest soils of Iceland

Marja Maljanen[1], Heli Yli-Moijala[1], Bjarni Didrik Sigurdsson[2], Christina Biasi[1]

[1]University of Eastern Finland, Department of Environmental and Biological Sciences, P.O.Box 1627, Finland
[2]Agricultural University of Iceland, Keldnaholt, 112 Reykjavik, Iceland

*Correspondence to*: **Marja Maljanen (marja.maljanen@uef.fi)**





**Abstract.** Carbon dioxide ($CO_2$) efflux and $\delta^{13}C$ in $CO_2$ were measured along a natural geothermal soil temperature ($T_s$)

gradient in upland Sitka spruce forest soil in a volcanic area in Iceland in July 2014 and 2016. The gradient that reaches from ambient soil temperature up to 40°C warming at 10 cm depth was originally formed in May 2008, following a major earthquake. The $CO_2$ efflux from the forest floor was measured using the static chamber method. In addition, subsurface soil $CO_2$ concentrations and $\delta^{13}C$ values of $CO_2$ were studied. In summer 2014, soil surface $CO_2$ efflux increased steadily with increasing soil temperature across a temperature gradient of 40ºC (from 260 to 3900 mg m$^{-2}$ h$^{-1}$). In 2016 the trend had changed;

the maximum $CO_2$ efflux (2100 mg m$^{-2}$ h$^{-1}$) was measured at 20ºC $T_s$ warming and a similar nonlinear trend was observed in soil $CO_2$ concentrations in 2016. The $^{13}C$ isotope analysis of $CO_2$ suggested that a proportion of the $CO_2$ emitted from the warmer plots was geothermally derived. The plot with the highest geothermal source was different in 2014 and 2016, which explained the shift in the temperature dependence of the total $CO_2$ efflux. Our study showed that a significant amount of $CO_2$ emitted from the higher warming levels of geothermal temperature gradients can have non-biotic origin and this has to be

taken into account when measuring respiration fluxes on such volcanic sites.



## 1 Introduction

Most of the soil organic carbon is stored at northern latitudes, particularly in the northern permafrost regions and boreal forest soils. Climate warming is almost twice as large in the North as the global average during the recent decades (IPCC, 2013),

with a temperature increase of up to +5 °C predicted for 2100. Consequently, there is high risk of increasing the release of soil carbon (C) due to microbial activity, primarily in the form of carbon dioxide ($CO_2$), to the atmosphere in the future (Crowther et al., 2016). To study responses of northern ecosystems to climate warming, often experimental approaches are used. Long-term ecosystem manipulations of temperature are rare because of the logistical and financial challenges of experimental warming at this scale (Kayler et al., 2015; De Boeck et al., 2015). Warming experiments may be quite artificial and also

introduce unintended artefacts. All this can be overcome by natural temperature gradients within small distance, which are available e.g. in geothermal areas (O'Gorman et al., 2014). There, the effects of soil warming on the ecosystem can be studied without the confounding effects of manipulative warming and divergent transect approaches (Sigurdsson et al., 2016). Geothermal activity can remain stable for many years, making it possible to investigate long-term warming effects, but major tectonic events can also create new hotspots, exposing previously unwarmed ecosystems to higher temperatures and enabling

to study recent (short-term) temperature responses (O'Gorman et al., 2014). Such natural soil temperature gradients can be found e.g. in geothermal systems in southwest Iceland.

The major aim of our study was to investigate changes in $CO_2$ efflux rates along soil temperature gradients to predict the effects of future soil warming on $CO_2$ effluxes of terrestrial ecosystems, particularly forest soils. Our original hypothesis was that a

significant warming will accelerate soil microbial heterotrophic processes, leading to enhanced $CO_2$ effluxes with temperature. First indirect support for this hypothesis comes from a study by Poeplau et al. (2019) showing a strong warming-induced depletion of carbon concentration in the soils of these geothermal areas. However, soil microbial respiration rates were, contrary to the aforementioned hypothesis, reduced in these warmed soil as shown in recent laboratory incubation experiments (Marañón-Jiménez et al., 2018) most likely due to the reduced carbon contents in warmed soils and a subsequent decline in

microbial biomass; however, mass-specific respiration rates increased with warming (Walker et al., 2018). Temperature effects on soil respiration are thus highly complex and they need to be studied in the field to fully elucidate the interactions of plants, soil, and climatic variation.

However, in volcanic active areas it is known that also some geothermal $CO_2$ can be emitted from the underlying volcanic

system (Ármansson, 2018; Stefánsson, 2017; Rey et al., 2015; Bia et al., 2014; Fridriksson et al., 2008; Klusman et al., 2000), which could possibly confound measurements of *in situ* soil respiration. Geothermal $CO_2$ effluxes are receiving a lot of attention in studies on geology and volcanology (Ármansson, 2018), but they have not been exhaustively explored in environmental studies aiming at elucidating temperature effects on respiration rates. To study temperature responses of net ecosystem exchange, ecosystem respiration or soil respiration in geothermal areas, such non-biogenic $CO_2$ effluxes need to be



included and disentangled from biogenic $CO_2$ production. Here we quantified the impact of the geothermal $CO_2$ efflux on
overall soil-derived $CO_2$ efflux, in order to be able to elucidate temperature impacts on biological respiration in geothermally
warmed soils of Iceland. Since $CO_2$ from biological respiration is isotopically highly distinct from $CO_2$ derived from non-
biotic sources, including geothermal $CO_2$ (e.g. Caliro et al., 2007; Chiodini et al., 2010; Tassi et al., 2012; Biasi et al., 2008),
we used a stable isotope approach ($\delta^{13}C$) to separate the different flux components.

## 2 Methods

### 2.1 Study site

The study site is located in southwest Iceland, in the surroundings of the village Hveragerði (64.008°N, 21.178°W), on land
owned by the Agricultural University of Iceland. In 2004-2014. The area had a mean annual air temperature of 5.2 °C and a
mean annual precipitation of 1431 mm (Icelandic Met Office, IMO). The growing season normally starts in May and ends in
late August. The soil type at the study sites is Brown Andosol (Arnalds, 2015), with relatively high pH (5.5-7.0) and large soil
water retention capacity (O'Gorman et al., 2014; Sigurdsson et al. 2016).

On the 29th of May, 2008, a major earthquake (magnitude 6.3 on the Richter scale) occurred in southwest Iceland (Halldórsson
and Sigbjörnsson, 2009), where typically ca. 70-100 years pass between such large earthquake episodes in this region. The
2008 earthquake caused large structural damages to infrastructures and affected geothermal systems close to its epicenter. One
such geothermal system moved from its previous location to a new and previously unwarmed area (Þorbjörnsson et al., 2009),
and the new belowground geothermal channels within the bedrock resulted in soil temperature increases in the soil above. The
soil temperature elevation measured at 10 cm soil depth reached >50 °C where the channels are closest to the surface
(O'Gorman et al., 2014).


The "ForHot" research network (www.forhot.is) was established in 2011 to bring scientists together to study how changes in
soil temperature affect various ecosystem processes (Sigurdsson et al., 2016; Kayler et al., 2015; O'Gorman et al., 2014). The
present study was conducted in an area warmed after 2008 earthquake during growing seasons 2014 and 2016. The site is a
mature Sitka spruce (*Picea sitchensis)* forest, planted in 1966. More information about the site conditions can be found in
Sigurdsson et al. (2016)

### 2.2 $CO_2$ efflux measurements with chambers

The $CO_2$ effluxes were measured using the opaque static chamber method (Maljanen et al., 2017). The measurements were
made along the temperature gradient in June 2014 and were repeated in July 2016. There were six gas flux sampling plots each
year (Table 1). The sampling plots were located both outside the warmed area and within it at different elevated soil





temperatures ($T_s$) (see Table 1). The plots were named according the warming levels measured in 2012 with site code FN and temperature elevation as +X °C, as described in earlier study by Maljanen et al. (2017). The number (+X) indicates soil warming at depth of 10 cm. No deeper soil temperatures were measured in this study. Due to unexpected changes in the temperature gradient between the years one sampling plot was different in 2014 (FN +10) than in 2016 (FN +6).

Three replicate chambers were used on each measurement plot. The metal flux chambers (ø = 26 cm, h = 30 cm) had a hole in the top for a sampling line and for a capillary line to avoid pressure effect. Prior to sampling the sharp edge at the bottom of the chamber was twisted 3-5 cm into soil and the top opening was sealed with a rubber septum. Plants were not removed from the soil surface before gas sampling, however, there were no plants in plots FN+1, FN+2 and FN+6, and only few in FN+0. In plot FN+10, vascular plants covered about 30% of the surface, in plot FN+20 there were only mosses and in plot FN+40 all

mosses were dead in 2016, whereas in 2014 there were still some living ones. A total of five to six gas samples (30 ml) were collected between 5 to 66 min after installing the sealed chamber. Within 4 h, the samples were injected into 12 ml Labco pre-evacuated vials (Labco Exetainer®) for gas analysis at University of Eastern Finland (UEF). Samples were analyzed within two weeks for $CO_2$ concentration and $\delta^{13}C$ values for $CO_2$. Soil temperatures at 5 and 10 cm were manually measured next to the chambers at each sampling time.

### 2.3 Soil gas and hot spring gas sampling

Concentrations of $CO_2$ in soil air were measured at the sampling plots simultaneously with the gas flux measurements in July 2014 and July 2016. Gas samples of 20 ml were taken with a stainless steel sampling probe (ø = 3 mm, l = 40 cm) at three soil depths; 5, 10 and 20 cm in 2014 and at depths of 10, 20, 30 and 40 cm in 2016. Samples of $CO_2$ were treated and analysed as described above.


Samples to measure the $\delta^{13}C$ values of the parent fluid or the abiotic/geothermal source were taken on April 9, 2017, from hot spring vents near the study site (see Supplement Table 1). Infrared gas analyzer (EGM-3) was used to find vents within the ForHot area that had high $CO_2$ concentrations in the steam. Where possible, a chamber (described earlier) was used to isolate hot air coming up in hot spring and air was sucked into the syringe through a hole on the top of it. Labco Exetainers ® were

emptied two times with a 50 ml syringe and then filled with a gas sample. During sampling the air temperature was +2 °C, sunny and wind 2-4 m s$^{-1}$ from N.  Soil temperature at 10 cm in non-warmed soils was 1-2 °C.

### 2.4 $CO_2$ concentration analysis

Concentrations of $CO_2$ were determined with a gas chromatograph (Agilent 6890N, Agilent Technologies, USA), equipped with an autosampler (Gilson, USA) and thermal conductivity detector (TC). Compressed air, containing 386 µl l$^{-1}$ of $CO_2$, was


used for daily calibration. The gas flux rates were calculated from the linear increase or decrease in the gas concentration with time in the headspace of the chamber (Maljanen et al., 2017).

### 2.5 Isotope analysis of $CO_2$ ($\delta^{13}CO_2$)

Analysis of $\delta^{13}C$ in $CO_2$ from chamber or soil gas measurements were carried out at UEF. Subsamples (1 ml) were taken from each gas vials and were injected into 12 ml vials filled with $N_2$ and then analyzed with a gas chromatography (GC) system

coupled to an isotope-ratio mass spectrometer (GC-IRMS) (Delta XP[plus]; Thermo Finnigan, Bremen, Germany) as described in Biasi et al. (2008). Briefly, $CO_2$ was first concentrated in liquid nitrogen via a precon unit, and then separated on a GC column (Pora Plot Q; 30 m length). Then, $CO_2$ was transferred to the ion source of the IRMS via a Conflow IV interface and open split unit. Helium was used as a carrier gas at a constant flow rate of 1 ml min[-1]. The $\delta^{13}C$ of $CO_2$ samples from hotsprings was analyzed at the University of Vienna (UNIVIE) by using a headspace gas sampler (Gas-Bench II, Thermo Fisher, Bremen,

Germany) coupled to an IRMS (Delta V Advantage, Thermo Fisher, Bremen, Germany). The results of the $\delta^{13}CO_2$ analysis are expressed as $\delta$ (in per mill) according to the following formula (1):

$$\delta = \left(\frac{R\ sample}{R\ Standard} - 1\right) x\ 1000, \tag{1}$$

where R sample for C is the $^{13}C/^{12}C$ ratio of the sample and R standard is the $^{13}C/^{12}C$ ratio of Vienna Pee Dee Belemnite

(VPDB). The precision of the $\delta^{13}C$ values was about 0.20‰, respectively, as determined from ten measurements of internal working standards (synthetic air with 300 ppm $CO_2$, Air Liquide, Finland) analyzed together with the samples in each run. The absolute accuracy of the analysis was determined by injecting at least three aliquots of calibrated reference gas (99.995 vol.-% purity; Air Liquide, Finland) along with each sample analysis. At both UEF and UNIVIE the reference $CO_2$ gas of the IRMS is regularly calibrated against international calibration standards (IAEA-6, ISO-TOP gas standards (Air Liquide) with certified

$^{13}C$ concentrations) achieving long-term accuracy and precision of the results in both laboratories and guarantying comparable results. Calibration with IAEA-6 is done with an elemental-analyzer (Flash EA 1112 Series, Thermo Finnigan, Bremen, Germany) coupled to the IRMS mentioned above.

### 2.6 Soil and vegetation sampling and analysis

Soil samples (sampling depths 0-5 cm and 5-10 cm) were collected from all sites in June 2014 and in July 2016. Soil pH and

electric conductivity (EC) were measured from soil/water slurry (15 ml soil: 50 ml milliQ-$H_2O$) from homogenized and pooled samples. Gravimetric soil moisture was determined by drying the soil for 24 h at 105 ˚C. Soil C and N contents and $\delta^{13}C$ values from dried and homogenized samples were determined with the EA-IRMS system at UEF described above. The long-term precision of a quality control standard (wheat) was < 0.15‰ for C isotope analysis and < 0.9% (relative error) for elemental analysis. Isotope results are expressed relative to V-PDB as $\delta^{13}C_{V-PDB}$ as also described above.




Plant samples (*Agrostis stolonifera*) were picked nearby the plots, but not exactly from the same place where the gas fluxes were measured because the ground vegetation was scarce. In addition, there were no *Agrostis stolonifera* or any other grasses growing between plots FN+2 to FN+6. Plant samples were also collected outside the warmest plot FN+40 in the area where the trees were already dead and also nearby the coolest plot outside the forest. In addition, samples were collected between the

two warmest plots FN+20 and FN +40. After sampling the leaves of *Agrostis stolonifera* were dried at 40°C for 48h and they were grinded for analysis of %C, %N and $\delta^{13}C$ as described above.

**2.7 Calculation of $\delta^{13}C$ of sources and isotope mixing model**

The $\delta^{13}C$ of $CO_2$ emitted from the FN plots were calculated with the Keeling plot approach (Biasi et al., 2008; Keeling, 1958) using data on $CO_2$ concentration and $\delta^{13}CO_2$ values for each sampling time over the entire chamber closure (n=6). The Keeling

plot approach uses a linear regression on the measured variables ($\delta^{13}C$ and mixing ratios) to determine the end-member isotope value of the excess $CO_2$ relative to the background (air) value (Keeling, 1958). The end-member value ($\delta^{13}CO_2$ of $CO_2$ respired) was obtained by extrapolating the linear regression to zero (y-axis intercept or Keeling plot).

The relative contribution of geothermal vs. biogenic $CO_2$ was calculated with the two-pool isotope mixing model (2):

$$f = \frac{\delta - \delta_0}{\delta_1 - \delta_0},\tag{2}$$

where f is the fraction of the geothermal source in overall $CO_2$ emissions, $\delta$ is the $\delta^{13}C$ of $CO_2$ of the mixture ($CO_2$ emitted from FN plots), $\delta_1$ is the $\delta^{13}C$ of the geothermal source and $\delta_0$ is the $\delta^{13}C$ of biological respiration (Biasi et al., 2008). The $CO_2$ emissions rates from the geothermal source were calculated by multiplying f with overall $CO_2$ emission rates, and the biological respiration rates were obtained by the difference between overall and geothermal emissions. To estimate the $\delta^{13}C$ of the

geothermal source, data on $CO_2$ concentration and $\delta^{13}CO_2$ values of the gas samples taken from the hot-springs were used (**See Supplement Table 1**). Since the $CO_2$ in the soil gas sampled from the hot-springs could have been affected by gas mixing between atmospheric and soil gases, we also applied the Keeling plot approach to obtain the $\delta^{13}C$ of the original source gas (**See supplement Fig. 1**). We assumed here that $CO_2$ in the hot springs would reflect a mixture of only two sources (atmospheric and geothermal $CO_2$). The $\delta^{13}C$ of biological respiration was represented by a typical end-member value of

characteristic C3-type plants (-28‰). This assumption was supported by measured data on $\delta^{13}C$ values of plants and soil in the study area (see below). Isoerror version 4.1. (Phillips and Gregg, 2001) was used to determine standard error of f, taking into account the standard deviation of the $\delta^{13}C$ values of the mixture and each contributing source, respectively. A 95% confidence interval was used to determine if the f values were significant from zero or 1, respectively.



We also proceeded to plot Keeling plots and calculated $\delta^{13}C$ of $CO_2$ from the soil gas profiles of the FN plots. The Keeling plot here violates the basic assumption of this method that only two sources contribute to the mixture of $CO_2$, since there were three (atmospheric, biogenic and geothermal $CO_2$). Thus, we used the Keeling plot here merely to evaluate, qualitatively, whether any conclusions can be drawn with regard to sources of $CO_2$ in the soil profile and to compare the isotopic composition of soil gas with the one of $CO_2$ emitted from the surface.

**2.8 Statistical analyses**

Correlations between the gas flux rates (data not normally distributed) and $T_s$ were tested with non-parametric Spearman rank correlation tests. For the correlations between other soil variables, Pearson correlation tests were used (IBM SPSS statistics 25).

**3 Results**

**3.1 Soil properties**

The measured soil properties from depths of 0-5 cm and 5-10 cm are presented in **Table 1**. In 2014 soil temperature ($T_s$) at depth of 10 cm increased along the sampling transect up to 50.4 °C, whereas in 2016 the maximum temperature at the warmest plot was as high as 75 °C (**Table 1**). Thus, soil temperature had increased between years 2014 and 2016 at the warmest plots (FN+40, FN+20) but not in the originally cooler plots (FN+0, FN+1, FN+6).


Soil pH did not change significantly along the gradient, but soil electric conductivity (EC) was higher in the warmest plots in 2016 but not in 2014 samples (**Table 1**). The total C concentration in soil (**Table 1**) ranged between 17.3 and 1.1% and there was a decreasing trend with increasing soil temperature, except that the highest $CO_2$ concentrations were measured from the topsoil of plot FN+6 in 2014. The inorganic C concentration in this site was less than 2% of the total C (Marañón-Jiménez et al., 2018) and therefore more than 98% of the total C was in organic form. Soil total N concentration (range from 0.15-0.80%)
did not show any clear decreasing trend with temperature, except the lowest concentrations were measured from the warmest plot (FN+40). Soil C/N ratio in the top soil (0-5 cm) varied from 6.3 in the warmest plot (FN+40) to 26.8 in the slightly warmed plots (FN+1). The $\delta^{13}C$ value of the soil ranged between -28.91‰ and -26.67‰ with no significant difference between plots along the temperature gradient. However, soil $\delta^{13}C$ values were more negative in topsoil (average -28.1‰) than in the 5-10
cm layer (average -27.5‰) (see **Table 1**).



**Table 1.** Soil properties in top 0-5 and 5-10 cm at the study plots in 2014/2016. FN+0 is the unwarmed plot and the value +X shows the increase in soil temperature at depth of 10 cm.  Note that plot FN+2 was not sampled in 2016.


| Plot | T (ºC) | pH $_{H2O}$ | EC (µS cm$^{-1}$) | Tot C (%) | δ $^{13}$C in soil (‰) | N (%) | C:N |
|------|--------|-------------|-------------------|-----------|------------------------|-------|-----|
| *0-5 cm* | | | | | | | |
| FN+0 | 10.2/10.1 | 5.5/5.9 | 26/31 | 13.4/7.8 | -28.41/-27.64 | 0.61/0.48 | 22.1/21.0 |
| FN +1 | 10.7/10.7 | 6.0/6.4 | 27/20 | 10.7/14.2 | -27.89/-28.91 | 0.61/0.62 | 17.4/26.8 |
| FN +2 | 11.2/nd | 5.8/nd | 17/nd | 11.0/nd | -28.04/nd | 0.61/nd | 17.9/nd |
| FN +6 | 12.6/11.3 | 5.7/6.1 | 11/19 | 14.5/17.3 | -28.77/-28.68 | 0.71/0.80 | 20.5/23.2 |
| FN +10 | 13.1/18.1 | 5.8/6.0 | 15/36 | 10.3/11.7 | -27.69/-28.33 | 0.67/0.79 | 15.5/17.5 |
| FN +20 | 15.6/49.5 | 5.9/5.4 | 26/155 | 7.6/6.0 | -28.30/-28.33 | 0.57/0.49 | 13.2/14.2 |
| FN +40 | 42.5/62.6 | 6.9/5.6 | 29/259 | 3.4/3.8 | -27.05/-27.89 | 0.40/0.34 | 8.3/11.9 |
| *5-10cm* | | | | | | | |
| FN+0 | 10.2/10.2 | 5.7/6.1 | 31/32 | 7.8/9.0 | -27.13/-27.88 | 0.48/0.41 | 16.1/19.9 |
| FN +1 | 10.8/10.9 | 6.1/6.1 | 25/36 | 5.6/6.0 | -26.98/-27.92 | 0.46/0.36 | 12.2/15.9 |
| FN +2 | 11.3/nd | 5.6/nd | 12/nd | 7.9/nd | -27.33/nd | 0.62/nd | 12.7/nd |
| FN +6 | 12.3/11.7 | 5.9/5.8 | 12/34 | 8.8/8.3 | -26.67/-27.99 | 0.61/0.52 | 14.4/9.0 |
| FN +10 | 13.1/18.9 | 6.2/5.9 | 10/26 | 7.1/7.3 | -26.72/-28.12 | 0.58/0.44 | 12.4/5.1 |
| FN +20 | 17.9/54.5 | 6.1/5.8 | 21/69 | 5.5/2.8 | -27.02/-27.79 | 0.51/0.36 | 10.7/12.2 |
| FN +40 | 52.2/75.0 | 7.0/6.7 | 13/66 | 1.9/1.1 | -27.94/-28.26 | 0.30/0.15 | 6.3/8.9 |

**3.2 CO₂ efflux and δ $^{13}$C in CO₂ of the total CO₂ flux**

Overall $CO_2$ effluxes showed different patterns in the two measurement years. The total $CO_2$ efflux rates (350 mg $CO_2$ m$^{-2}$ h$^{-1}$) were similar between plots FN+0 and FN+6 in 2014, but increased then constantly up to the warmest plot, FN+40, where

the total $CO_2$ efflux reached its maximum value (>4000 mg $CO_2$ m$^{-2}$ h$^{-1}$). In 2016, $CO_2$ effluxes were slightly increasing with temperature in plots between FN+0 and FN+6 (from 170 to 300 mg $CO_2$ m$^{-2}$ h$^{-1}$), reached the maximum in plot FN+10 (2100 mg $CO_2$ m$^{-2}$ h$^{-1}$), and thereafter decreased in plot FN+40 down to 940 mg $CO_2$ m$^{-2}$ h$^{-1}$ (**Fig. 1**)

The δ$^{13}$C values of $CO_2$ differed along the temperature gradient and also between years (**Fig. 1**). In 2014, similar $^{13}$C values

were found for gases emitted from FN+0 and FN+2, where soil temperature differed only by ca. 3 °C. However, the $^{13}$C $CO_2$



values decreased significantly and progressively by nearly 7‰ in $CO_2$ emitted from the warmer plots (from -25.3‰ in FN+2 to -18.3‰ in FN+40). In 2016, the pattern in the observed $CO_2$ $\square^{13}C$ values was different; then the far lowest $\delta^{13}C$ of $CO_2$ was found in FN+10 plot (-5.32‰), while the values for FN+20 and FN+40 were higher (-11.7 and -14.2‰), but still significantly different from FN+0 (-26.7‰). There was also a progressive enrichment in the $\delta^{13}C$ of $CO_2$ (from -21.0 to -15.8‰) between

plots FN+1 and FN+6 in 2016. The $\delta^{13}C$ signal of the geothermal $CO_2$ measured from the nearby hot spring vents is shown in **supplement Table 1**. The $\delta^{13}C$ geothermal source value calculated using the Keeling plot approach for the geothermal $CO_2$ was -5.07±1.79‰ (**Supplement Fig. 1**).

### 3.3. $CO_2$ flux components: results of the isotope mixing model

The absolute amount of $CO_2$ emitted from the different flux components (biological and geothermal) was calculated with the

mass balance approach. The estimated geothermal component in the total soil $CO_2$ efflux increased progressively from a few percent (not significantly different to zero in FN+0), to almost 100% in FN+40 in 2014, with no significant difference between the first three plots (**Fig. 1**). In 2016, the proportion of geothermal $CO_2$ efflux increased progressively from a few percent (but not significantly different to zero percent) in FN+0, to about 30% in FN+1 to more than 90% in FN+10, and decreased thereafter down to 60% in FN+40 (**Fig. 2**), indicating that a spatial shift had taken place in the location of the maximum

geothermal outgassing between the two years.

In 2014 the biological $CO_2$ efflux rate remained at the same level on the three coolest plots, did not correlate with soil temperature within those plots and dropped about 70% in FN+20 plot compared to the coolest plots and was zero in the warmest FN+40 plot. In 2016 there was a different trend within the research area, the biological $CO_2$ efflux was at relatively low level,

and not significantly different between the sites in the coolest plots, but it was further dropping in F+10 plot and thereafter increasing again in the warmest plots (**Fig. 1**). In 2014 there was an increasing trend but not a significant correlation with both total $CO_2$ efflux and geological $CO_2$ efflux and soil temperature at depth of 10 cm. In 2016 there were no linear correlations between total $CO_2$ efflux rates and soil temperatures, again indicating that the spatial shift in outgassing was not mirrored with changes in geothermal temperature.

### 3.3 $CO_2$ concentrations in soil and $\delta^{13}C$ in $CO_2$

Soil air $CO_2$ concentrations, sampled from 5, 10 and 20 cm in 2014 and from 10, 20, 30 and 40 cm in 2016 soil depth differed significantly between plots and increased with depth (**Fig. 3**). In 2014 $CO_2$ concentrations also increased with increasing soil temperature and depth and the highest concentrations were measured from the warmest plot FN+40 (up to 180 000 µl l$^{-1}$) at depth of 20 cm (data not shown). In 2016 the $CO_2$ concentrations from the coolest plot (FN +0) were similar than in 2014

(from 1300 to 2200 µl l$^{-1}$) but the highest $CO_2$ concentration at depth of 20 was 70 000 µl l$^{-1}$, measured from plot FN+10. In plots FN+20 and FN+40 the soil $CO_2$ concentrations were lower than that in 2016 (**Fig. 3**). The $\delta^{13}CO_2$ values of soil air, which





were measured only in 2016, also differed significantly between FN+0 and the warmer plots (FN+10, FN+20 and FN+40). In general, changes followed the isotopic signal of $CO_2$ efflux at the surface, with highest $\delta^{13}CO_2$ of soil air measured at FN+10 (-6.03) and lowest at FN+0 (-23.8).

### 3.4. Plants

Plant samples collected from the gas sampling plots or close to those were analyzed for C, N and $\delta^{13}C$ and there was an increasing trend ($R^2 = 0.806$, $p = 0.006$) in plant tissue C concentrations with increasing soil temperature (**Supplement Table S2**). The total N concentrations were more variable and there was no significant temperature trend (**Supplement Table S2**). A clear difference was found on the $\delta^{13}C$ values. In the warmest plots, FN+20 and FN+40, the values were less negative than in cooler plots (**Fig. 4**). One outlier from this trend was plot FN+10, which was sampled only in 2016 because in 2014 no *Agrostis stolonifera* plants were found nearby the plot.

### 4. Discussion

Isotope results show clearly that non-biological $CO_2$ was emitted from the site, especially from the geothermally warmest plots but also, to a variable extent, from the more mildly warmed plots. In 2014 the contribution of geothermal $CO_2$ increased with increasing temperature and the $CO_2$ emitted from the warmest plot (FN+40) was almost totally geothermal origin (ca. 99 %). This highest measured geothermal flux (4 g $CO_2$ m$^{-2}$ h$^{-1}$) was in the range of that Ármannsson (2018) has reported for outgassing in geothermal areas in Iceland.

The geothermal emissions were not tightly connected to warmest temperatures in 2016. Then there was also a progressive increase in geothermal $CO_2$, but only between plots FN+0 and FN+10, corresponding to 6 to 98 % contribution of geothermal $CO_2$ efflux. Then the highest proportion of geothermal $CO_2$ efflux was not from the hottest plot, where it was only 60 %, but from FN+10. The $CO_2$ concentration in soil also showed also different trends in 2014 and 2016 as we could see from the surface $CO_2$ fluxes. The less negative $\delta^{13}C$ values from the plot FN+10 in 2016 confirmed that the geothermal source was strongest at that point whereas in 2014 it was strongest in plot FN+40. This indicates a shift in the spatial location of the outgassing and that it is not necessarily following the same pattern as the thermal diffusion from the underlying bedrock. Geothermal emissions can also occur at low temperatures as a result of outgassing of buried gas reservoir or due the presence of deep sources of fluids, common for the occurrence of metamorphic processes (Chiodini et al., 2010) and can be of episodic nature. However, the geothermal component in the total soil $CO_2$ efflux decreased quite regularly away from the point of maximum outgassing in both years: This may offer a practical way to factor out non-biogenic $CO_2$ fluxes during *in situ* flux measurements than having to simultaneously measure $^{13}CO_2$ signature of all measurement plots. Further studies are, however, needed on the temporal variation of the geothermal outgassing before such methods could be recommended.



The temporal variability of geothermal $CO_2$ efflux was most likely because the geothermal channels had changed most probably due to a minor earthquake (2.7 on Richter scale) on July 8, 2015 (Icelandic Met Office, IMO), causing changes in the location of largest geothermal $CO_2$ source. We noticed that there was a shift also in the higher end of the temperature gradient between the years 2014 and 2016.

Since the outcome of the isotope mixing model is dependent on the isotope values of the end-members, the impact and potential uncertainties arising from potentially different isotope signatures of the sources need to be discussed. The $\delta^{13}C$ values of $CO_2$ from the hot spring vents were in the range of those reported earlier from hydrothermal systems (Caliro et al., 2007), though relatively more depleted. Carbon dioxide from volcanic hydrothermal discharge areas usually have $\delta^{13}C$ value between 0.5 ‰ and -2 ‰, especially when the source of magmatic origin (Tassi et al., 2015). Only when the geothermal gases are characterized by more crustal $CO_2$, the $\delta^{13}C$ values can also be more negative, down to -11 ‰ (Tassi et al., 2016). In our case, the $\delta^{13}CO_2$ values were on average -5.07 ± 1.7 ‰, thus a mixture of both sources is likely. There could have been also some variability in isotope signatures of biological source since those can change with temperature, plant cover and $CO_2$ source (Bogue et al., 2019). In our case, the $\delta^{13}C$ of *Agrostis stolonifera* plant increased slightly at the highest temperatures, most likely due to re-assimilation of enriched $CO_2$ stemming from geothermal sources. However, this trend was not noticeable in the soil where the $\delta^{13}C$ values were relatively constant along the temperature gradient suggesting only minor impact on biological processes. Generally, due to the large differences in isotope values of the two sources, the results of the isotope mixing model was not very sensitive to changes in $\delta^{13}C$ of both end-members, and the accuracy of each source could have been determined with relatively high confidence.

After subtracting the abiotic proportion of $CO_2$ from overall $CO_2$ efflux, trends in biotic $CO_2$ effluxes can be discussed. However, temperature trends were difficult to elucidate here, since it was is not possible to differentiate heterotrophic respiration from autotrophic respiration in all plots; thus we are merely discussing in the following differences in biological respiration rates between plots and the interpretation of the data with respect to the temperature gradient has to be taken with caution. Biological $CO_2$ efflux was not increasing when soil temperature was increasing (up to +20 °C warming) along the gradient in 2014, it actually stayed at relatively constant level. Some of the effects could be due to changed belowground allocation of trees and changes in understory. Indeed amount of fine roots biomass sharply decreased at higher soil temperatures in the forest (Parts et al., 2019), which is in accordance our findings. Amount of understory was also lower at FN+1, FN+2 and FN+6 than at FN+0 and therefore could have partly contributed to the apparent decreased $CO_2$ efflux at the higher temperatures because of reduced autotrophic respiration from the understory aboveground parts. However, despite some understory growing in FN+20, biological respiration rates still decreased there. No measurements of fine-root biomass exist for FN+20 and FN+40, but there the tree overstory has died (O'Gorman et al., 2014). Positive effects of temperature on heterotrophic microbial activity might also have been counterbalanced by lower carbon content of the soil, leading to "apparent



acclimation" of $CO_2$ effluxes at the higher soil temperatures, as suggested also in other laboratory studies in ForHot project by Marañón-Jiménez et al. (2018) and Walker et al. (2018).

Still it was unexpected to see the *in situ* biological $CO_2$ efflux rates drop to almost zero in the warmest plot (FN+40) in 2014,
which may be also a result of unfavorable conditions (e.g. high soil $CO_2$ concentration, high temperature) for microbes in soil and therefore changes in microbial population (Oppermann et al., 2010). Unpublished studies on microbial responses at the site have indeed found a shift in the microbial composition at the highest temperatures, but still containing viable bacterial communities (James Weedon and Erland Baath; personal comm.), and similar results with warming have been found for soil bacteria and fungi in the nearby ForHot grassland sites (Radujkovic et al., 2018). It was noteworthy that in 2016 the lowest
biological emissions also shifted and coincided with the highest geological emissions at FN+10, which could have been the results of toxic effects of high geothermal soil $CO_2$ concentrations and possibly other gases in soil. However, the low biogenic $CO_2$ efflux rates at the points of the highest outgassing remain unexplained. More studies are obviously needed to get more detailed information on temperature responses from biological respiration in situ from geothermal areas.

As mentioned above, the $\delta^{13}C$ values in the plant samples taken from plots with the strongest geothermal $CO_2$ source in 2014 were moderately more enriched in 2016 than the samples from other plots. This could possibly be showing that substantial amounts of geothermal $CO_2$ can be taken up by the perennial *A. stolonifera* plants (Oppermann et al., 2010) in the earlier years when the highest outgassing was at FN+40 and these changes $\delta^{13}C$ values in plants could be used as an indicator for a strong geothermal source (Bowling et al., 2008; Bogue et al., 2019).


**Conclusion**

This study shows that the geothermal $CO_2$ emission may have a significant role in overall $CO_2$ efflux from geothermal areas and this should be taken into account when measuring net $CO_2$ emissions and planning and conducting isotope experiments on such sites. Precise partitioning of $CO_2$ efflux between its two main sources is needed, in order to get information on the
temperature response of *in situ* biological soil respiration. This is also important for calibrating soil C models for field $CO_2$ flux rates. There can be a large overestimation of the biogenic $CO_2$ efflux when the corresponding isotopic data are not considered. Source partitioning of $CO_2$ will be critical for field studies; laboratory studies are not impacted by the geothermal $CO_2$ sources from deep-origin (Marañón-Jiménez et al., 2018; Walker et al., 2018; Maljanen et al., 2018).

**Data availability**

Data from the study are available for collaborative use by anyone interested. Contact M. Maljanen (marja.maljanen@uef.fi).



**Author contribution**

MM, CB and BDS designed the experiments and MM and HYM carried out sampling from the study site. CB was responsible for the $\delta^{13}C$ analysis and calculations and MM for the other analysis. MM prepared the manuscript with contributions from all
co-authors.

**Competing interests**

The authors declare that they have no conflict of interest.

**Acknowledgements**

Hanne Vainikainen, Jaana Rissanen, Simo Jokinen and Tatiana Trubnikova are thanked for assisting in the laboratory. Andreas
Richter is thanked for the gas analysis at UNIVIE. This work contributes to the COST ES1308 ClimMani, the Nordic CAR-ES and the ForHot (www.forhot.is) network projects and the Icelandic Research Fund project 163272-053. The Icelandic-Finnish Cultural Foundation is thanked for travel grants to MM. We also want to acknowledge the staff at the Reykir campus of Agricultural University of Iceland for great logistic support. The study was further supported by strategic funding of the University of Eastern Finland (project FiWER granted to CB).

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






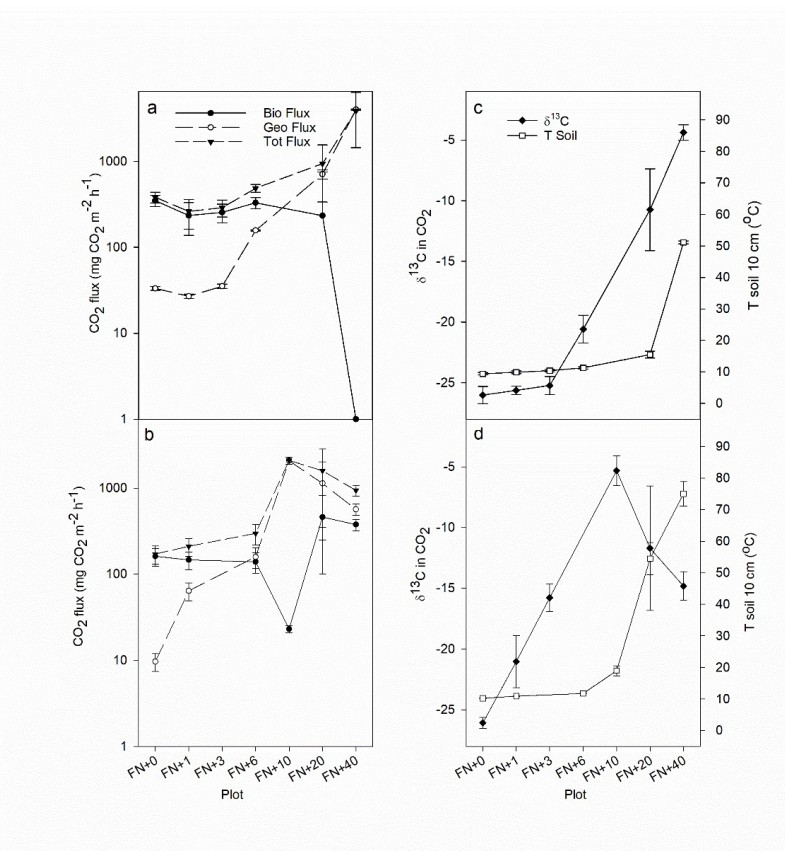

**Fig. 1. On the left the total CO₂ efflux, biological CO₂ efflux and geological CO₂ efflux in 2014 (a) and 2016 (b). On the right the corresponding δ¹³C in CO₂ values with soil temperature at depth of 10 cm in 2014 (c) and 2016 (d). Error bars show the standard deviation (n=3). Note the logarithmic scale for CO₂ efflux.**

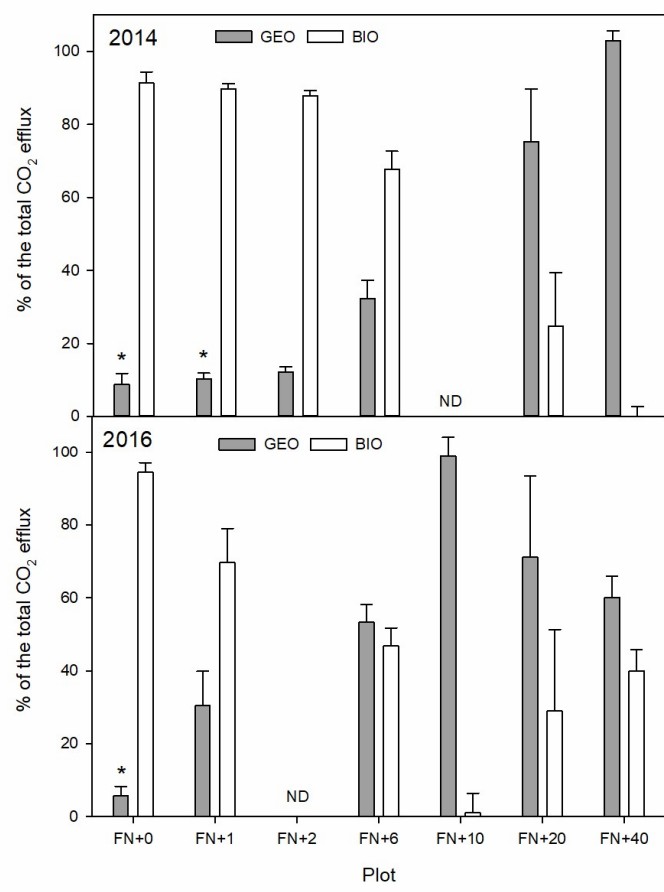


**Fig. 2.** The average (± stdev) CO₂ effluxes measured from three replicate chambers from each plot during sampling campaigns in July 2014 and June 2016. The bars are showing the percentage of biological or geothermal origin, calculated based on $\delta^{13}C$ isotope analysis of emitted CO₂. ND = the plot was not measured. * = percentage values are not statistically significant from zero.


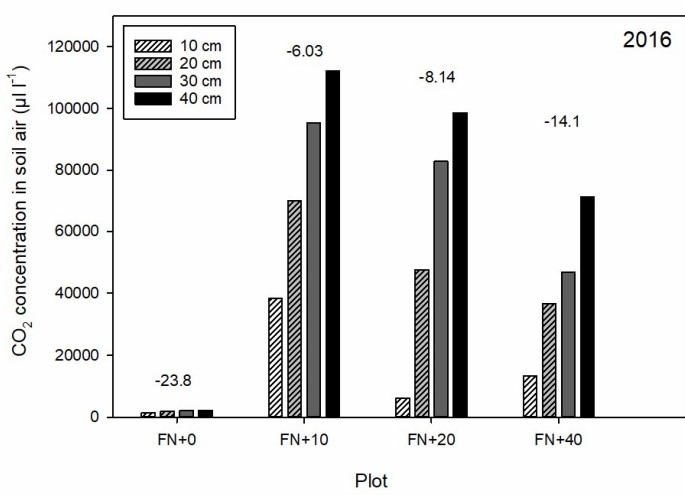

**Fig. 3. The average (n=3) concentration of CO₂ in soil air sampled from depths of 10, 20, 30 and 40 cm in June 2016 with the corresponding δ¹³C values for CO₂ (mixture of biological and geothermal) calculated with Keeling plot method.**

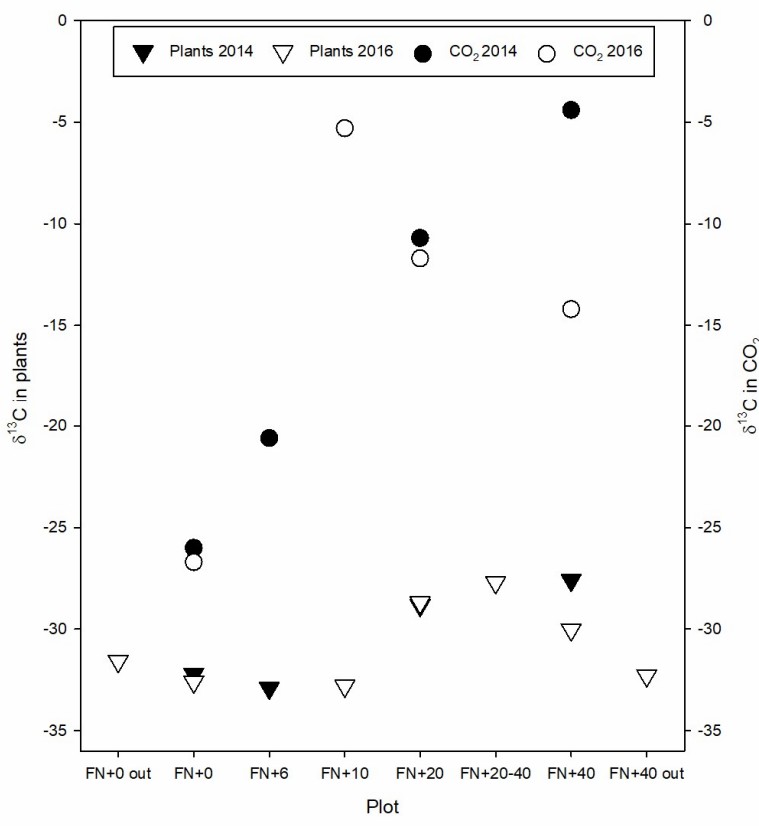

**Fig. 4. The δ¹³C values in plants collected from the study plots, outside the plots (out) and between FN+20 and FN+40 (FN+20-40) plots plotted with δ¹³C values in CO₂ efflux measured with the chambers and calculated with Keeling plot approach (n = 3). The plant samples consisted of several leaves of *Agrostis stolonifera*, which were mixed and pooled before analysis.**
