# Peer review of "Abiotic CO2 sources confound interpretation of temperature responses of in situ respiration in geothermally warmed forest soils of Iceland"

_Biogeosciences, 2019_

## Referee Comment (RC1) · Anonymous Referee #1 · 30 Jul 2019

This manuscript describes the contribution of abiotic (geothermal) CO2 source to the total CO2 emission from geothermally warmed forest soils of Iceland. The topic is relevant to BGD, the MS is well written and the techniques employed are appropriate. Some moments should be taken into account before final publication of MS: 1. I advise using in the title 'biotic CO2 efflux' instead of 'respiration' 2. There are 3 weak methodological points in the work. - The first is a very small repetition of CO2 emission measurements, which is insufficient for obtaining truthful results due to the very high spatial and high variability of soil CO2 fluxes; - The second is the difference in vegetation and its density in the study plots where the CO2 fluxes were measured. Since the authors did not remove the vegetation, this is a significant moment that

could affect the CO fluxes from soils. The comparison between plots, in this case, cannot be considered legitimate. The convincing explanations on these issues are required; - The third is the absence of any statistical analyses of soil and CO2-flux data. 3. Due to the region studied is very exotic it would be nice to include more information on the relevance of this study for other regions. It may be analyses of the temperature sensitivity (e.g. Q10 values) of biotic components of total CO2 emission using the data for plots FN+0, FN+1, FN+2, FN+6, and FN+10 plots. 4. Some specific comments: - in Fig. 1, the lines for designating total and geo- CO2-fluxes are very similar. Use, please, more contrasting symbols for lines; - Table 1 in Supplement: Include, please, mean and SE in this table instead of the individual measurements; - Fig 1 (Supplements): Change please the scale (1/concentration), using the 10-3 for scaling. The Figure will be more readable.

Please also note the supplement to this comment:
https://www.biogeosciences-discuss.net/bg-2019-213/bg-2019-213-RC1-supplement.pdf

**Supplement:**

**Review for MS:**

Title: Abiotic CO2 sources confound interpretation of temperature responses of in situ respiration in geothermally warmed forest soils of Iceland
Author(s): Marja Maljanen et al.
MS No.: bg-2019-213
MS Type: Research article

| Principal criteria | Excellent (1) | Good (2) | Fair (3) | Poor (4) |
|---|---|---|---|---|
| **Scientific significance:** Does the manuscript represent a substantial contribution to scientific progress within the scope of Biogeosciences (substantial new concepts, ideas, methods, or data)? | | | X | |
| **Scientific quality:** Are the scientific approach and applied methods valid? Are the results discussed in an appropriate and balanced way (consideration of related work, including appropriate references)? | | X | | |
| **Presentation quality:** Are the scientific results and conclusions presented in a clear, concise, and well-structured way (number and quality of figures/tables, appropriate use of English language)? | | X | | |

This manuscript describes the contribution of abiotic (geothermal) CO2 source to the total CO2 emission from geothermally warmed forest soils of Iceland. The topic is relevant to BGD, the MS is well written and the techniques employed are appropriate.

Some moments should be taken into account before final publication of MS:

1. I advise to use in the title 'biotic CO2 efflux' instead of 'respiration'
2. There are 3 weak methodological points in the work.
   - The first is a very small repetition of CO2 emission measurements, which is insufficient for obtaining truthful results due to the very high spatial and high variability of soil CO2 fluxes;
   - The second is the difference in vegetation and its density in the study plots where the CO2 fluxes were measured. Since the authors did not remove the vegetation, this is a significant moment that could affect the CO fluxes from soils. The comparison between plots in this case cannot be considered legitimate. The convincing explanations on these issues are required;
   - The third is the absence of any statistical analyses of soil and CO2-flux data.
3. Due to the region studied is very exotic it would be nice to include more information on relevance of this study for other regions. It may be analysis of the temperature sensitivity (e.g. Q10 values) of biotic components of total CO2 emission using the data for plots FN+0, FN+1, FN+2, FN+6, and FN+10 plots.
4. Some specific comments:
   - in Fig. 1, the lines for designating total and geo- CO2-fluxes are very similar. Use, please, more contrasting symbols for lines;

- Table 1 in Supplement: Include, please, mean and SE in this table instead of the individual measurements;
- Fig 1 (Supplements): Change please the scale (1/concentration), using the $10^{-3}$ for scaling. The Figure will be more readable.

---

## Referee Comment (RC2) · Werner Eugster (Referee) · 25 Aug 2019

This is an interesting topic and fits into Biogeosciences. However, before acceptance I recommend a few improvements of the paper (rather moderate revisions). The language level is OK, but definitely needs the standard language check done by Biogeosciences.

Although the topic is very interesting, I struggled in some parts with the lack of care in the wording. For example I first did not realize that "soil $\delta^{13}$C" is actually not what I expected (i.e., the $\delta^{13}$C signature of the organic material in the soil), but is jargon for $\delta^{13}$C in soil CO$_2$, which is a completely different thing.

Another point is how the author deal with missing data. The justifications are not really convincing. For example, it remained unclear to me why there are no 2014 data from FN+10 presented in Figure 4. The text on lines 255–256 sais that there was no sampling in 2014 because no *Agrostis stolonifera* plants were found. Well, this only relates to the plant tissue $\delta^{13}$C, but it does not explain why there was no soil CO$_2$ found on which an isotopic analysis could have been carried out.

And the third point is: even if everything is fine, I expected some discussion why the plant tissue in Figure 4 is below the range of the mixing model. To me it appears that only the FN+20, FN+20–40 and FN+40 plant tissue signatures are in the range determined for soil respiration (−26.67 to −28.91‰ according to line 198).

Finally, I have some critique on the experimental design: the authors discuss temporal variabilty of geothermal CO$_2$ effluxes, but their sampling only presents two snapshots, one from one (!) campaign in June 2014, and one in July 2016. Thus, no variablity can be deduced in a scientifically defensible way from two snapshots. Why did the authors not e.g. add iButton temperature loggers to convince me about the average temperature conditions? It remains obscure how they defined the FN+X classification without such data, and Table 1 actually shows that the deviation of the snapshot temperatures from the planned +X °C temperature class can be substantial.

Thus, let me summarize my critique in the hope that this helps to improve the manuscript in the revisions:

1. **Terminology.** Please say explicitly if you mean air in the soil and not the soil itself. Also when providing mass units, it is essential to know whether this is mass of C or mass of CO$_2$ (see details below). Also the term "unwarmed" sounds like "cooled" to me. Later in the text you use the term "non-warmed" which I understood probably correctly. Do not use synonyms in scientific texts, this is confusing, and be more specific and clear in your choice of wordings for special terms. Here I definitely would not use the term "unwarmed".

2. **Missing data.** The absence of data points (missing data) is not fully described. I see the aspect of reproducibility in such issues: as a reviewer I need to know whether the authors do not present the data points because they did not like them, or whether they made a sound an serious, objective and reproducible decision about screening out data. Here only partial information is given and what is presented is only partially convincing the reviewer. Please specify all information about missing data. For example, on lines 146–151 you say that there was no *Agrostis stolonifera* found between plots FN+2 to FN+6 (my interpretation of such a statement is that FN+2, and FN+6 had the plants, but between the two plots you did not find this plant, but the presentation in Figure 4 suggest that FN+2 and FN+6 did not have the plant either). But only on line 255 (at the end of the Results section) do you inform me that also FN+10 did not have this plant in 2014, an information that was not provided in the Methods section. I would expect all this information clearly presented in the Methods section, and then in the Discussion section you could discuss how the missing data might have influenced your interpretation. I also expected a statement why there are no soil $CO_2$ $\delta^{13}C$ measurements presented in Figure 4 for FN+0 out and FN+40 out. Moreover, you have FN+1 in Table 1, but in Figure 4 this plot is quietly removed. Why? You must be more clear what you show and what you hide (and why).

3. **Plant tissue isotopic signatures.** In Figure 4 most of your triangles are **below** the lower end-member (soil respiration) of your mixing model. Only the data from FN+20, FN+20–40 and FN+40 seem to be in agreement with 100% soil respiration (which is questionable as well) but I would expect a discussion why all the tissue values are so much lower and cannot be explained with your mixing model.

4. **Experimental design.** Please provide the key information in the Methods section how the plot selection and assignment of the temperature deviations was done. Was this also only based on snapshot data? Is there no way to have a

better statistical information on the longer-term mean temperatures of all plots? Also specify the exact dates of sampling, not only the month, this is essential information if someone wants to consider weather conditions (you do not lose a single word on this) when you did the sampling. I fear that all interpretations are somewhat uncertain since we do not know whether you had comparable weather conditions in June 2014 and July 2016 or whether I as a reviewer should be concerned that one snapshot had much lower air temperatures than the other.

5. **What was the expectation?** You did not phrase any science questions or hypotheses, which makes it difficult for the reader to know what you expected and why I might not have expected the same. For example: Figure 2 shows almost a reversed gradient of the share of geogenic $CO_2$ flux from what you probably expected (an implicit information deduced from how you present your data). But I wondered: what did you actually expect? When do you think the temperature deviation is high and when is it low? I would expect that if there are no macropores in the soil then the surface soil temperatures might be higher than if there are macropores and cracks where the geogenic heat and $CO_2$ an reach the soil surface more easily and quickly than under absence of cracks and macropores, and thus the temperature of the topsoil does not pick up as much of that heat as if it were purely molecular heat transfer. Thus I wondered whether the 2016 conditions might reflect a change in macropores – and on line 279 you indeed mention a minor earthquake, but do not follow up on this important information in the discussion. Thus: could it not be that in 2016 FN+10, FN+20 and FN+40 were affected by getting more cracks/macropores, and thus the temperature increase from 2014 to 2016 under presence of cracks and macropores might just reflect a reversed picture? If there is not a good reason for not having monitored the soil temperatures over a longer period (e.g. with iButtons), you should definitely address this flaw of the experimental design in a critical manner in your discussion.

**DETAILS** (with line numbers)

11: "formed" does not sound correct here. Maybe established, created, resulted, . . .

14–15: I assume the mass units relate to $CO_2$, not C; thus write $mg\,CO_2\,m^{-2}\,h^{-1}$. Personally, I am from a community that uses $\mu mol\,m^{-2}\,s^{-1}$, not mass units, then there is no need to specify such details.

16: it is not a $^{13}C$ analysis, but an analysis of the ratio of $^{13}C/^{12}C$. Avoid jargon!

17: what does the word "different" exactly refer to? Does it mean that the plot with highest geothermal source had a different source strength in 2014 compared to 2016? Or do you think that it was not the same plot in 2014 and 2016 that had the highest source strength? In the latter case you should not use "the plot" as the subject of the phrase and rather write something like "It was not the same plot that had the highest geothermal source strenght in 2014 and 2016" (but if this is what you want to say: why not say FN+X had the highest . . . in 2014, whereas in 2016 the highest . . . were observed in FN+Y?).

63: "In 2004-2014." is not a correct sentence. Connect with the following sentence.

97: this confuses me: you write that the sample analysis was done at University of Eastern Finland, but in the acknowledgments you thank Andreas Richter for gas analysis at UNIVIE. What is now the correct information? The two statements are in stark conflict! Also specify how the samples were transported to the (correct) laboratory, and what precautions had been taken to minimize transportation artefacts.

98: Soil temperature measurements: specify what instrument/method you used. This is lacking completely.

125 (and elsewhere): no space between $\delta$ and $^{13}C$

127: no $\times 1000$ needed in the equation. $\delta$ values are a ratio which can be presented as a ratio or as a percentage or as permils. But this is not a unit conversion and thus the

$\times 1000$ is an error in the equation (although people still use this – if you add a permil sign then it is no longer an error, but not best practice).

135: find an English word for "guarantying"

150: the past of "grind" is "ground"

155: linear regression (if you mean ordinary least-squares regression OLS) is not the correct best practice for Keeling plots. Use orthogonal regression (statistically perfect would be standardized major axis regression, but other orthogonal regressions also tend to be OK for the task). Maybe this solves some issues discussed under point #3 above, OLS is always flatter than orthogonal regressions since OLS does not consider uncertainty in the values ploted on the x-axis).

172: please only use the term standard error as defined in statistics; if you bootstrap as with Phillips and Gregg (2001) then the correct term is "uncertainty". Note that standard error is only relating to normally distributed data, and it is $SD/\sqrt{n}$, whereas bootstrapping is distribution-independent.

174: wording should rather be "significantly different from zero"

175–179: please reword! The Keeling plot does not violate your assumptions, it is just the other way round: you as authors violate the assumptions made by the Keeling plot! Don't blame Keeling for this!

175–179: the Phillips and Gregg (2001) model deals with underdetermined systems; why do you not use this approach here instead of the Keeling plot, if you consider three sources/sinks? This is not clear to me.

181–183: you used Pearson's product-moment correlation. Did you check whether your data are really normally distributed? You never mentioned that you carefully tested your data for model assumptions. If the data are not normally distributed, then you should use Spearman's correlation coefficient instead.

198–199: clarify that you are not writing about $\delta^{13}C$ of the soil, but of $\delta^{13}C$ in $CO_2$ in the air in the soil. This is quite a different thing! (2 occurrences)

Table 1: I prefer manuscripts with tables and figures at the end. If you want to place them in the text then they should appear near the first reference. Table 1 is referred to first on line 84, but you placed it between lines 200 and 206; this neither helps the typesetter nor the reviewer.

Table 1 caption: please specify in Methods why FN+2 was not sampled in 2016. Or did I miss this?

Table 1 heading: say "in soil $CO_2$" not only "in soil"

214–217: you seem to have changed your font for $\delta$ but forgot to export the font to the PDF (3 occurrences). Please rectify (and check your PDF more carefully for errors related to your word processing program)

222: there is a rule to round reported figures to signifcant digits. Here it would be $-5 \pm 2\text{‰}$ or at at best $-5.1 \pm 1.8\text{‰}$ I am however not sure whether the isotope community respects this rule. In any case: make sure you present the same numbers with the same accuracy (and improved rounding) both on line 222 and line 289.

224: this is an error: you're not writing about absolute amounts of $CO_2$ in what follows, but about **relative** amounts.

238–239: see my comments in major point #5 related to this statement here.

249: when reporting values that are not dimensionless please add the units after the figures.

252: did you check for normal distribution of the data both on the x- and y-axes? If not, please report and if they are not normally distributed, then use the correct statistics for trend testing (Mann-Kendall trend test)

255–256: explanation not satisfactory (see introductory comments above)

273–276: you use the term "point of maximum outgassing" here, but you have not defined what this means. As it stands, the whole statement remains obscure and not really understandable to me. Also the sentence that follows is inaccessible to me, maybe because of the lack of definition and explanation of "point of maximum outgassing". This needs some rewording and additional explicit explanation.

295–296: don't use the term "accuracy" if you do not have an absolute reference; use the term "precision" instead. And I do not agree that there is high confidence, unless you used an orthogonal regression approach (and explain which one you think gives the highest precision of the regression slope in this application). In **R** the lmodel2 package provides major axis and standardized major axis regression and provides references. The Editor in charge (Dan Yakir) might give you an authorative answer, which approach is currently considered best practice here. Get his advice.

306–308: probably your interpretation is OK, but without soil moisture information it is difficult to make a sound judgement. Why did you not also measure soil moisture? Is the soils wet no matter what the temperature is? In most cases respiration at high temperatures decresase because of the negative correlation with soil moisture. Is this what you expect and explain here? Please provide more insight and explicitly state your understanding of the soil moisture aspect. I think that if the trees die then this might be a result of dry soils, but without your explicit information it is blind guessing on my side (at best).

Figure 1: for me it is confusing that you have FN+0, FN+1, FN+3, FN+6, ... here, but in Table 1 you have FN+2 in place of FN+3. Is this a simple typo or something that must be added to Methods? And why is there no soil temperature available for FN+3? There is no word about this (see my point #2 above). Also it is unclear why no soil temperature was measured at FN+10 in 2014, although in Table 1 you have values for what I assume to be soil temperatures (the caption however does not say what $T$ really is . . . ).

Figure 3: please add ‰ after the isotope ratio, or divide them by 1000 (then without ‰).

Figure 4: needs more information in Methods. It remains obscure what FN+0 out and FN+40 out actually show, since you explained that also in the other plots you collected the plants **outsite** the plot. So my assumption is that FN+0 out is more outsite than the FN+0 is, but this is quite fuzzy. Please add the necessary information to your Methods section. And please modify the title of your y-axis at right. It is not any $CO_2$ that you show here (and it also is not soil $CO_2$), it seems to be the $CO_2$ in your flux chamber. My suggestion thus is to label this axis "$\delta^{13}C$ in $CO_2$ efflux".

Signed review: Werner Eugster

---

## Author Comment (AC1) · 10 Sep 2019

1. I advise to use in the title 'biotic CO2 efflux' instead of 'respiration' Reply: We will change respiration to biotic CO2 as suggested by the reviewer

2. There are 3 weak methodological points in the work. - The first is a very small repetition of CO2 emission measurements, which is insufficient for obtaining truthful results due to the very high spatial and high variability of soil CO2 fluxes; Reply: This is true, the study plots were small and only limited amounts of samples could be taken. However, we wanted to focus our study on the transect and wanted to include several plots differing in temperature, also we wanted to take at the same time the isotope samples,

therefore we compromised on the number of replicates to make this practically feasible. Still, we have high confidence in our results, since the three replicates taken from each plot were very similar and the changes along the gradient were seen very well. A similar sampling approach with three replicates was also chosen in Maljanen et al. (2017), and the results matched ours. The main differences in $CO_2$ fluxes were seen between the sites (and not between the replicates) along the gradient. This manuscript focuses on interpreting these differences, by mainly studying if there are any (and how much) non-biological $CO_2$ emissions.

- The second is the difference in vegetation and its density in the study plots where the $CO_2$ fluxes were measured. Since the authors did not remove the vegetation, this is a significant moment that could affect the CO fluxes from soils. The comparison between plots in this case cannot be considered legitimate. The convincing explanations on these issues are required; Reply: It is true that vegetation affects the $CO_2$ flux. These marked sampling plots were very small and also many other experiments were going on by other scientist there and therefore we were not allowed to remove vegetation or otherwise disturb the study plots. We will add this explanation to a revised version of the manuscript. However, as mentioned above, the present study focuses on the existence of non-biological $CO_2$ emission, and tries to find correlations between environmental factors and non-biological fluxes in order to be able to correct for these abiotic fluxes in future studies (which was proved impossible). Therefore, the discussion on the temperature sensitivity of the biological fluxes has here minor importance. In the revised version of the manuscript, we would clarify this issue.

- The third is the absence of any statistical analyses of soil and $CO_2$-flux data. 3. Due to the region studied is very exotic it would be nice to include more information on relevance of this study for other regions. It may be analysis of the temperature sensitivity (e.g. Q10 values) of biotic components of total $CO_2$ emission using the data for plots FN+0, FN+1, FN+2, FN+6, and FN+10 plots. Reply: First, as the reviewer also mentioned in his/her pervious comment, the results on the biological respiration

are impacted by vegetation and thus Q10 values, unfortunately, are not very useful. But the main aim was not here to study the biological respiration but the amount of abiotic sources of CO2, by using isotope analysis. This is very important to know if these sites are used for studying the effects of soil warming on biological processes (e.g. soil respiration). We tried to correlate the abiotic CO2 fluxes with temperature and other environmental parameters but could not find any correlation. It seems that these emissions are random and not easy to predict, thus isotope analysis are always necessary, making it difficult to generalize the results. We mention that temperature gradient studies from volcanic areas need to consider the two components of the CO2 fluxes, however, we would add some more text to the these concluding remarks in a revised version.

4. Some specific comments: - in Fig. 1, the lines for designating total and geo- CO2-fluxes are very similar. Use, please, more contrasting symbols for lines; Reply: We will change the lines/symbols - Table 1 in Supplement: Include, please, mean and SE in this table instead of the individual measurements; - Fig 1 (Supplements): Change please the scale (1/concentration), using the 10-3 for scaling. The Figure will be more readable. Reply: We will do these changes to make the figures more readable.

---

## Author Comment (AC2) · 10 Sep 2019

This is an interesting topic and fits into Biogeosciences. However, before acceptance I recommend a few improvements of the paper (rather moderate revisions). The language level is OK, but definitely needs the standard language check done by Biogeosciences. Although the topic is very interesting, I struggled in some parts with the lack of care in the wording. For example I first did not realize that "soil $\delta$13C" is actually not what I expected (i.e., the $\delta$13C signature of the organic material in the soil), but is jargon for $\delta$13C in soil CO2, which is a completely different thing. C1 Reply: Thanks for the comment, we will revise the manuscript and take special care to the wording, in

order to make clear what is meant e.g. with $\delta$13C (e.g. $\delta$13CO2 of soil gas) and other phrases.

Another point is how the author deal with missing data. The justifications are not really convincing. For example, it remained unclear to me why there are no 2014 data from FN+10 presented in Figure 4. The text on lines 255–256 sais that there was no sampling in 2014 because no Agrostis stolonifera plants were found. Well, this only relates to the plant tissue $\delta$13C, but it does not explain why there was no soil CO2 found on which an isotopic analysis could have been carried out. Reply: It is true that there are some "missing" data, but the case is not that we were removing some data or hiding them. It was possible to take only limited number of samples from the small plots. Soil gas isotopes were not measured in 2014. These soil gas isotope analysis were included in 2016 after we had realized how interesting the flux results were from 2014. The reason why we changed one sampling plot between 2014 and 2016 were the changes in soil temperatures and we wanted to cover better the whole temperature gradient. These CO2 sampling plots are not connected to plant sampling. The reason why there were no plant samples from certain plots was simply because there were not any plants growing on those plots. We will clarify this issue in a revised version of the manuscript.

And the third point is: even if everything is fine, I expected some discussion why the plant tissue in Figure 4 is below the range of the mixing model. To me it appears that only the FN+20, FN+20–40 and FN+40 plant tissue signatures are in the range determined for soil respiration (–26.67 to –28.91‰ according to line 198). Reply: That is a good point and needs to be clarified, indeed. Plant tissue is always a little bit more negative in $\delta$13C (depleted in 13C) than soil and particularly soil CO2 even if there is no abiotic CO2. The $\delta$13C of soil is also always increasing with depth in arable soils. This is because of a small (but progressive) discrimination against 13C during heterotrophic respiration and/or (not yet clear) increased contribution of 13C-enriched microbially derived C with depth. That is why we used also $\delta$13C of biological respiration for

the mixing model, and not soil $\delta$13C values of soil. We will clarify this point and add respective references (e.g.  Boström et al., Oecologia 153(1):89-98).

Finally, I have some critique on the experimental design: the authors discuss temporal variabilty of geothermal CO2 effluxes, but their sampling only presents two snapshots, one from one (!) campaign in June 2014, and one in July 2016. Thus, no variablity can be deduced in a scientifically defensible way from two snapshots. Why did the authors not e.g. add iButton temperature loggers to convince me about the average temperature conditions? It remains obscure how they defined the FN+X classification without such data, and Table 1 actually shows that the deviation of the snapshot temperatures from the planned +X ◦C temperature class can be substantial. Reply: Originally the plots were named based on temperature data measured with data loggers in 2012 by AUI, explained in Maljanen et al. 2017. For these short campaigns in 2014 and 2016 we did not install any ibuttons. We did our best in showing spatial and temporal variability, however, surely particularly the temporal design could have been improved. In future studies, we will take samples over multiple years and compare the abiotic fluxes (and particularly impact on biotic fluxes). We expect also significant variability within a season, and plan to study this in an upcoming experiment. However, this was out of the scope of the current manuscript which presents, to our knowledge, the first study showing the significant influence of abiotic CO2 fluxes on overall CO2 fluxes in studies using geothermal gradients to simulate climate change. The discussion on the temporal variability can be, however, toned down in a revised version.

Thus, let me summarize my critique in the hope that this helps to improve the manuscript in the revisions: 1. Terminology. Please say explicitly if you mean air in the soil and not the soil itself. Also when providing massunits ,it is essential to know whether this is mass of C or mass of CO2 (see details below). Also the term "unwarmed" sounds like "cooled" to me. Later in the text you use the term "non-warmed" which I understood probably correctly. Do not use synonyms in scientific texts, this

is confusing, and be more speciïñĄc and clear in your choice of wordings for special terms. Here I deïñĄnitely would not use the term "unwarmed". Reply: We will consider all the comments with respect to terminology as suggested by the reviewer and hope this makes the manuscript clearer.

2. Missing data. The absence of data points (missing data) is not fully described. I see the aspect of reproducibility in such issues: as a reviewer I need to know whether the authors do not present the data points because they did not like them, or whether they made a sound an serious, objective and reproducible decision about screening out data. Here only partial information is given and what is presented is only partially convincing the reviewer. Please specify all information about missing data. For example, on lines 146–151 you say that there was no Agrostis stolonifera found between plotsFN+2toFN+6(my interpretation of such a statement is that FN+2, and FN+6 had the plants, but between the two plots you did not ïñĄnd this plant, but the presentation in Figure 4 suggest that FN+2 and FN+6 did not have the plant either). But only on line 255 (at the end of the Results section) do you inform me that also FN+10 did not have this plant in 2014, an information that was not provided in the Methods section. I would expect all this information clearly presented in the Methods section, and then in the Discussion section you could discuss how the missing data might have inïñĆuenced your interpretation. I also expected a statement why there are no soil CO2 $\delta$13C measurements presented in Figure 4 for FN+0 out and FN+40 out. Moreover, you have FN+1 in Table 1, but in Figure 4 this plot is quietly removed. Why? You must be more clear what you show and what you hide (and why). Reply: We will be much clearer in presenting data in a revised version of the manuscript and apologize that this was confusing. As mention above, there are no data hidden or removed on purpose, the choice of sampling site and sampled data was bot arbitrary and there is a plausible explanation for any lack of data, and we will give it convincingly in a revision. E.g., there were no soil CO2 $\delta$13C measurements presented for FN+0 out and FN+40 out because they were outside the original plots and only plant samples were collected. Again, we were able to take only limited amount of soil/gas samples and transport them in reasonable

time in Finland for analysis. Gas samples were not taken from plot F1 for the same reason. No data from those plots were removed or hidden.

3. Plant tissue isotopic signatures. In Figure 4 most of your triangles are below the lower end-member (soil respiration) of your mixing model. Only the data from FN+20, FN+20–40 and FN+40 seem to be in agreement with 100% soil respiration (which is questionable as well) but I would expect a discussion why all the tissue values are so much lower and cannot be explained with your mixing model. Reply: see answer to the reviewer's comment above

4. Experimental design. Please provide the key information in the Methods section how the plot selection and assignment of the temperature deviations was done. Was this also only based on snapshot data? Is there no way to have a better statistical information on the longer-term temperatures of all plots? Also specify the exact dates of sampling, not only the month, this is essential information if someone wants to consider weather conditions (you do not lose a single word on this) when you did the sampling. I fear that all interpretations are somewhat uncertain since we do not know whether you had comparable weather conditions in June 2014 and July 2016 or whether I as a reviewer should be concerned that one snapshot had much lower air temperatures than the other. 5. What was the expectation? You did not phrase any science questions or hypotheses, which makes it difficult for the reader to know what you expected and why I might not have expected the same. For example: Figure 2 shows almost a reversed gradient of the share of geogenic CO2 flux from what you probably expected (an implicit information deduced from how you present your data). But I wondered: what did you actually expect? When do you think the temperature deviation is high and when is it low? I would expect that if there are no macropores in the soil then the surface soil temperatures might be higher than if there are macropores and cracks where the geogenic heat and CO2 an reach the soil surface more easily and quickly than under absence of cracks and macropores, and thus the temperature of the topsoil does not pick up as much of that heat as if it were purely molecular heat transfer. Thus

I wondered whether the 2016 conditions might reflect a change in macropores – and on line 279 you indeed mention a minor earthquake, but do not follow up on this important information in the discussion. Thus: could it not be that in 2016 FN+10, FN+20 and FN+40 were affected by getting more cracks/macropores, and thus the temperature increase from 2014 to 2016 under presence of cracks and macropores might just reflect a reversed picture? If there is not a good reason for not having monitored the soil temperatures over a longer period (e.g. with iButtons), you should definitely address this flaw of the experimental design in a critical manner in your discussion. Reply: As mentioned above, originally the plots were named based on temperature data measured with data loggers in 2012 by AUI, explained in Maljanen et al. 2017. Sampling dates (9.-11.6.2014 and 10.-11.7.2016) and the mean air temperature during those sampling periods (which was 14 oC during both campaigns) will be added in the manuscript. The aim of this study was not to follow the change in the soil temperature or behavior of the geothermal system at the site but to track the different sources of $CO_2$ and to show the possible risks of disturbance by geological $CO_2$ when using geothermal gradients in warming experiments. We agree that we can only speculate about the possible reasons behind the temporal and spatial variability of the abiotic $CO_2$ fluxes. In our view, any detailed discussion on this topic would be beyond the scope of this manuscript.

DETAILS (with line numbers) 11: "formed" does not sound correct here. Maybe established, created, resulted, ... 14–15: I assume the mass units relate to $CO_2$, not C; thus write $mgCO_2$ m$-2$ h$-1$. Personally, I am from a community that uses $\mu$molm$-2$ s$-1$, not mass units, then there is no need to specify such details. 16: it is not a 13C analysis, but an analysis of the ratio of 13C/12C. Avoid jargon! 17: what does the word "different" exactly refer to? Does it mean that the plot with highest geothermal source had a different source strength in 2014 compared to 2016? Or do you think that it was not the same plot in 2014 and 2016 that had the highest source strength? In the latter case you should not use "the plot" as the subject of the phrase and rather write something like "It was not the same plot that had the highest geothermal source strenght in

2014 and 2016" (but if this is what you want to say: why not say FN+X had the highest ... in 2014, whereas in 2016 the highest ... were observed in FN+Y?). Reply: Thank you for your comment, we will clarify this phrasing.

63: "In 2004-2014." is not a correct sentence. Connect with the following sentence. 97: this confuses me: you write that the sample analysis was done at University of Eastern Finland, but in the acknowledgments you thank Andreas Richter for gas analysis at UNIVIE. What is now the correct information? The two statements are in stark conflict! Reply: The isotope analysis were done at two different institutes, samples from the study plots were analyzed at UEF and samples from geothermal sources, geothermal vents outside the plots (data in supplement) were done at UNIVIE. We will clarify this part.

Also specify how the samples were transported tot he(correct)laboratory, and what precautions had be taken to minimize transportation artefacts. Reply: Samples were transported by plane within two days after sampling in the luggage of the researcher, which was the fastest and safest way to get them analyzed. With the knowledge from our previous experiments the transportation will not cause any significant damage. Sample transportation information will be added in the manuscript.

98: Soil temperature measurements: specify what instrument/method you used. This is lacking completely. Reply: Name and type of the thermometer will be added.

125 (and elsewhere): no space between $\delta$ and 13C 127: no$\times$1000 needed in the equation. $\delta$ values are a ratio which can be presented as a ratio or as a percentage or as permils. But this is not a unit conversion and thus the$\times$1000 is an error in the equation (although people still use this – if you add a permil sign then it is no longer an error, but not best practice). 135: find an English word for "guarantying" 150: the past of "grind" is "ground" Reply: Thanks for these notes, we will change these.

155: linear regression (if you mean ordinary least-squares regression OLS) is not the correct best practice for Keeling plots. Use orthogonal regression (statistically perfect

would be standardized major axis regression, but other orthogonal regressions also tend to be OK for the task). Maybe this solves some issues discussed under point #3 above, OLS is always flatter than orthogonal regressions since OLS does not consider uncertainty in the values ploted on the x-axis). Reply: This is new to us, and we appreciate the reviewer's comments. We will try the orthogonal regressions in a revised version of the manuscript. However, linear correlation was only accepted if R2 was > 0.95 (which indicates good fit, and which was true in most case). Thus, we doubt that the main message will change, but for sake of correctness the orthogonal regressions will be calculated.

172: please only use the term standard error as defined in statistics; if you bootstrap as with Phillips and Gregg (2001) then the correct term is "uncertainty". Note that standard error is only relating to normally distributed data, and it is SD/$\sqrt{n}$, whereas bootstrapping is distribution-independent. 174: wording should rather be "significantly different from zero" Reply: Thanks for clarification. We will use the term "uncertainty" if data were not normally distributed and the term "standard error" if data were normally distributed in a revised version of the manuscript.

175–179: please reword! The Keeling plot does not violate your assumptions, it is just the other way round: you as authors violate the assumptions made by the Keeling plot! Don't blame Keeling for this! Reply: True, we will rephrase according to the reviewer's suggestion. E.g. "Since Keeling plot assumptions were not met, we. . ..."

175–179: the Phillips and Gregg (2001) model deals with underdetermined systems; why do you not use this approach here instead of the Keeling plot, if you consider three sources/sinks? This is not clear to me. Reply: This is a good point. We will use the Phillips and Gregg (2001) model to calculate source contributions to soil CO2 by using the isotope data in a revised version of the manuscript. From first attempts in running this model, however, it looks like the main outcome does not change. Importantly, the data on d13CO2 from soil gas were sampled to complement the surface CO2 flux data and to corroborate our interpretation that abiotic CO2 contributes to surface CO2

emissions in geothermal gradients.

181–183: you used Pearson's product-moment correlation. Did you check whether your data are really normally distributed? You never mentioned that you carefully tested your data for model assumptions. If the data are not normally distributed, then you should use Spearman's correlation coefiňĄcient instead. Reply: Spearman correlation was used when data was tested to be not normally distributed.

198–199: clarify that you are not writing about $\delta$13C of the soil, but of $\delta$13C in CO2 in the air in the soil. This is quite a different thing! (2 occurrences) Reply: We will make sure and clarify the terminology in a revised version of the manuscript. Thanks for your careful comment!

Table 1: I prefer manuscripts with tables and iňĄgures at the end. If you want to place them in the text then they should appear near the iňĄrst reference. Table 1 is referred to iňĄrst on line 84, but you placed it between lines 200 and 206; this neither helps the typesetter nor the reviewer. Table 1 caption: please specify in Methods why FN+2 was not sampled in 2016. Or did I miss this? Reply: This will be clarified in methods and Table 1 will be moved close to line 84.

Table 1 heading: say "in soil CO2" not only "in soil" 214–217: you seem to have changed your font for $\delta$ but forgot to export the font to the PDF (3 occurrences). Please rectify (and check your PDF more carefully for errors related to your word processing program) Reply: We are sorry for these copy-paste errors. Those will be taken care of in a revised version of the manuscript.

222: there is a rule to round reported iňĄgures to signifcant digits. Here it would be –5$\pm$2‰ or at at best –5.1$\pm$1.8‰ I am however not sure whether the isotope community respects this rule. In any case: make sure you present the same numbers with the same accuracy (and improved rounding) both on line 222 and line 289. 224: this is an error: you're not writing about absolute amounts of CO2 in what follows, but about relative amounts. Reply: Thanks for your comments, we will correct these in a revised

version of the manuscript

238–239: see my comments in major point #5 related to this statement here. Reply: See our answers above

249: when reporting values that are not dimensionless please add the units after the figures. 252: did you check for normal distribution of the data both on the x- and y-axes? If not, please report and if they are not normally distributed, then use the correct statistics for trend testing (Mann-Kendall trend test) 255–256: explanation not satisfactory (see introductory comments above) Reply: These will be done, thanks! All explanations can and will be added to full satisfaction (see answers above)

273–276: you use the term "point of maximum outgassing" here, but you have not defined what this means. As it stands, the whole statement remains obscure and not really understandable to me. Also the sentence that follows is inaccessible to me, maybe because of the lack of definition and explanation of "point of maximum outgassing". This needs some rewording and additional explicit explanation. Reply: Maybe the word "point" is misleading, we were referring to the plot which had the maximum abiotic $CO_2$ emissions. We will change the wording.

295–296: don't use the term "accuracy" if you do not have an absolute reference; use the term "precision" instead. And I do not agree that there is high confidence, unless you used an orthogonal regression approach (and explain which one you think gives the highest precision of the regression slope in this application). In R the lmodel2 package provides major axis and standardized major axis regression and provides references. The Editor in charge (Dan Yakir) might give you an authorative answer, which approach is currently considered best practice here. Get his advice. Reply: We will first replace the term "accuracy" with "precision" and then try different approaches to calculate the Keeling plot to make sure the most accurate source value is obtained. We know the editor Dan Yakir and will ask for his advice concerning that matter. However, we do not think that the main outcome of the study will be influence by different calculations of the Keeling plot since differences we experienced were very large and linear fit was good. But for sake of correctness, however, the suggested corrections will be implemented and we are thankful for the comments.

306–308: probably your interpretation is OK, but without soil moisture information it is difficult to make a sound judgement. Why did you not also measure soil moisture? Reply: Gravimetric soil moisture measured in 2014 is shown in Maljanen et al. 2017 supplement. We did also measure gravimetric moisture in 2016, but the results are rather similar, there were only minor differences between plots, except FN+40, which had much lower water content which obviously affects biological $CO_2$ emission. This data can be added in the manuscript.

Is the soils wet no matter what the temperature is? In most cases respiration at high temperatures decresase because of the negative correlation with soil moisture. Is this what you expect and explain here? Please provide more insight and explicitly state your understanding of the soil moisture aspect. I think that if the trees die then this might be a result of dry soils, but without your explicit information it is blind guessing on my side (at best). Figure 1: for me it is confusing that you have FN+0, FN+1, FN+3, FN+6, ... here, but in Table 1 you have FN+2 in place of FN+3. Is this a simple typo or something that must be added to Methods? And why is there no soil temperature available for FN+3? There is no word about this (see my point #2 above). Also it is unclear why no soil temperature was measured at FN+10 in 2014, although in Table 1 you have values for what I assume to be soil temperatures (the caption however does not say what T really is ...). Reply: These will be clarified in the manuscript. See also replies to earlier comments.

Figure 3: please add ‰ after the isotope ratio, or divide them by 1000 (then without ‰. Figure 4: needs more information in Methods. It remains obscure what FN+0 out and FN+40 out actually show, since you explained that also in the other plots you collected the plants outsite the plot. So my assumption is that FN+0 out is more outsite than the FN+0 is, but this is quite fuzzy. Please add the necessary information to your Methods

section. And please modify the title of your y-axis at right. It is not any CO2 that you show here (and it also is not soil CO2), it seems to be the CO2 in your flux chamber. My suggestion thus is to label this axis "$\delta$13C in CO2 efflux". Reply: All true, we will follow the reviewer's suggestions.

―――――――――――――